# Baseline cardiometabolic profiles and SARS-CoV-2 infection in the UK Biobank

**Ryan J. Scalsky**[1], **Yi-Ju Chen**[2], **Karan Desai**[2], **Jeffery R. O'Connell**[2], **James A. Perry**[2]*, **Charles C. Hong**[2]*

1 Medical Scientist Training Program, University of Maryland School of Medicine, Baltimore, Maryland, United States of America, 2 Department of Medicine, University of Maryland School of Medicine, Baltimore, Maryland, United States of America

* JPerry@som.umaryland.edu (JAP); Charles.hong@som.umaryland.edu (CCH)

## Abstract

### Background

SARS-CoV-2 is a rapidly spreading coronavirus responsible for the Covid-19 pandemic, which is characterized by severe respiratory infection. Many factors have been identified as risk factors for SARS-CoV-2, with much early attention being paid to body mass index (BMI), which is a well-known cardiometabolic risk factor.

### Objective

This study seeks to examine the impact of additional baseline cardiometabolic risk factors including high density lipoprotein-cholesterol (HDL-C), low density lipoprotein-cholesterol (LDL-C), Apolipoprotein A-I (ApoA-I), Apolipoprotein B (ApoB), triglycerides, hemoglobin A1c (HbA1c) and diabetes on the odds of testing positive for SARS-CoV-2 in UK Biobank (UKB) study participants.

### Methods

We examined the effect of BMI, lipid profiles, diabetes and alcohol intake on the odds of testing positive for SARS-Cov-2 among 9,005 UKB participants tested for SARS-CoV-2 from March 16 through July 14, 2020. Odds ratios and 95% confidence intervals were computed using logistic regression adjusted for age, sex and ancestry.

### Results

Higher BMI, Type II diabetes and HbA1c were associated with increased SARS-CoV-2 odds ($p < 0.05$) while HDL-C and ApoA-I were associated with decreased odds ($p < 0.001$). Though the effect of BMI, Type II diabetes and HbA1c were eliminated when HDL-C was controlled, the effect of HDL-C remained significant when BMI was controlled for. LDL-C, ApoB and triglyceride levels were not found to be significantly associated with increased odds.

**Data Availability Statement:** All relevant data are within the paper and its Supporting information files.

**Funding:** This research was conducted using the UK Biobank Resource under Application Number 49852. This work was supported by National

Institute of General Medical Sciences (NIGMS) 5T32GM092237-10 to RS, NIGMS R01GM118557, National Heart, Lung, and Blood Institute (NHLBI) R01HL135129 to CCH, and NHLBI 1U01HL137181 to JP. The funders had no role in study design, data collection and analysis, decision to publish, or preparation of the manuscript.

**Competing interests:** The authors have declared that no competing interests exist.

## Conclusion

Elevated HDL-C and ApoA-I levels were associated with reduced odds of testing positive for SARS-CoV-2, while higher BMI, type II diabetes and HbA1c were associated with increased odds. The effects of BMI, type II diabetes and HbA1c levels were no longer significant after controlling for HDL-C, suggesting that these effects may be mediated in part through regulation of HDL-C levels. In summary, our study suggests that baseline HDL-C level may be useful for stratifying SARS-CoV-2 infection risk and corroborates the emerging picture that HDL-C may confer protection against sepsis in general and SARS-CoV-2 in particular.

## Introduction

Since early December 2019, when the first cases of the severe acute respiratory syndrome coronavirus 2 (SARS-CoV-2 or Covid-19) were identified in Wuhan, China, nearly 13 million individuals have tested positive for the virus [1]. Researchers have rapidly attempted to define the clinical characteristics associated with increased risk of becoming infected with SARS-CoV-2 to improve our understanding and clinical management of this pandemic. Early data from across the globe have identified pre-existing cardiovascular disease and obesity as risk factors associated with acquiring SARS-CoV-2 [2–8]. Curiously, early data from China also found that hypolipidemia and declining HDL-C at the time of acute COVID19 illness was associated with disease severity [9, 10]. There remains limited research on how an individual's baseline cardiometabolic profile, specifically lipid levels, affect one's risk for contracting the virus. This has received particular attention as the known viral entry mechanism for SARS-CoV-1, a closely related virus responsible for the 2003 SARS outbreak in China, has been shown, in preliminary in-vitro studies, to be cholesterol-dependent [11]. In this paper we analyze the association of a positive SARS-CoV-2 test with an individual's lipid profile in the UK Biobank resource.

## Methods

The UK Biobank resource began releasing SARS-CoV-2 test results in April 2020 to approved researchers. Data collected by the UK Biobank resource covers a wide variety of areas that researchers apply for access to. This data was not collected for this study but rather this study leveraged available collected data. This data was consented for by participants through UK Biobank protocol and was fully de-identified prior to approved researchers accessing it. Full details on these test results are available online [12]. Using test results released on July 14, 2020, we classified subjects testing positive for SARS-CoV-2 as cases. If multiple tests were performed, we classified a subject as a case if any test gave a positive result, based on the rationale that false positives are less likely than false negatives. Those with only negative test results were classified as controls. Tests were initially conducted in hospital settings in individuals who presented with respiratory symptoms. From April 27th onward, testing was expanded to include community clinics and all non-elective patients admitted overnight, including those who were asymptomatic. 70.1% of the 9,005 subjects (cases and controls) were inpatient when the sample was taken, 74.1% of 7,497 controls were inpatient when the sample was taken and 50.3% of 1,508 cases were inpatient when the sample was taken. The vast majority of samples for testing were obtained by nose/throat swabs and samples were analyzed for SARS-CoV-2 RNA via PCR.

The association analysis was performed with Plink2 [13] using logistic regression. The binary outcome variable of "SARS-CoV-2 test status" (cases tested positive, controls tested negative) was run against a series of independent variables which included continuous, categorical, and binary ICD10 data supplied by the UK Biobank. The data for continuous and categorical traits were collected when subjects were enrolled into the UK Biobank (2006–2010). Serum was collected for analysis of LDL-C, HDL-C, ApoA-I, ApoB and triglycerides. LDL-C was analyzed by enzymatic selective protection, HDL-C was analyzed by enzyme immunoinhibition, ApoA-I and ApoB were analyzed by immunoturbidimetric analysis and triglycerides were measured by GPO-POD. All analyses were completed using the AU5800 by Beckman Coulter. Height and weight were measured, and BMI was computed from these values. Impedance BMI was measured by bioelectrical impedance using the Tanita BC418MA body composition analyzer. Additional details on collection and analysis of biomarkers (e.g. LDL-C, HDL-C, ApoA-I) are available from the UK Biobank website [14]. ICD10 diagnostic codes are current for all subjects through October 2019. We also grouped the ICD10 code data into Phecodes in order to increase statistical power [15]. The analysis included covariates of sex, age, and principal components (PCs) 1 through 4 to adjust for ancestry. Principal component analysis, a standard technique used in statistical genetics, generates a dataset of PCs (typically 10) that can be used as covariates to correct for population stratification (i.e. differences in ancestry) [16]. PCs provided by the UK Biobank, which were computed from the cohort's genotypes, were used. Our preliminary analysis showed that only the first 4 PCs were significant at $p < 0.05$ and thus we included only PC1-4 as covariates.

Our analysis yielded odds ratios (OR) and 95% confidence intervals (CI) for each trait tested against the "SARS-CoV-2 test status". An OR greater than 1.0 indicates increased odds of a SARS-CoV-2 positive test compared to the controls. An OR less than 1.0 indicates decreased odds. For continuous phenotypes, the OR indicates the increased odds (for OR > 1.0) or decreased odds (for OR < 1.0) per standard deviation increase in the continuous phenotype.

## Results

### Demographics

This dataset includes 1,508 cases and 7,497 controls for a total of 9,005 subjects. Prior to the association analysis we compared the cases and controls for differences in sex, ancestry and age. Significant differences were found between the sex (p-value = $1.3 \times 10^{-3}$; Table 1), ancestry (p-value = $1.1 \times 10^{-15}$; Table 1) and age (p-value = $3.4 \times 10^{-8}$; Table 1) of cases and controls.

**Table 1. Demographics—Sex, ancestry and age.**

|  | N | Male (%) | Female (%) | White (%) | Non-white (%) | Age (SD) |
|---|---|---|---|---|---|---|
| Cases | 1,508 | 796 (52.8) | 712 (47.2) | 1312 (87.0) | 196 (13.0) | 67.39 (9.22) |
| Controls | 7,497 | 3,618 (48.3) | 3,879 (51.7) | 6980 (93.1) | 517 (6.9) | 68.81 (8.38) |
| All | 9,005 | 4,414 (49.0) | 4,591 (51.0) | 8292 (92.1) | 713 (7.9) | 68.57 (8.54) |
| % positive | 16.7% | 18.0% | 15.5% | 15.8% | 27.5% | N/A |
| p-value | | $1.3 \times 10^{-3}$ | | $1.1 \times 10^{-15}$ | | $3.4 \times 10^{-8}$ |

*N/Male/Female/Non-white/White indicate number of subjects. Age is the mean age as of 2020, SD is standard deviation. P-values are from chi-squared test for sex and ancestry, and t-test for age, comparing cases and controls. White ancestry includes subjects self-reporting as White, British, Irish, or "Any other white background". Non-white ancestry includes all other self-report categories.

**Table 2. Effect of body mass index.**

| Trait | Covariates | N | Cases/Controls | Mean (S.D.) | Odds Ratio | 95% Confidence Interval | p-value |
|---|---|---|---|---|---|---|---|
| BMI | Age, Sex, 4pc | 8,939 | 1,496/7,443 | 28.29 (5.27) | 1.12 | 1.06–1.18 | **$6.14 \times 10^{-5}$** |
| BMI | Age, Sex, 4pc, HDL-C | 7,764 | 1,280/6,484 | 28.31 (5.26) | 1.06 | 0.995–1.13 | 0.071 |
| Impedance BMI | Age, Sex, 4pc | 8,739 | 1,463/7,276 | 28.29 (5.26) | 1.12 | 1.06–1.18 | **$8.54 \times 10^{-5}$** |
| Impedance BMI | Age, Sex, 4pc, HDL-C | 7,590 | 1,252/6,338 | 28.30 (5.25) | 1.06 | 0.994–1.13 | 0.077 |

*Body Mass Index (BMI) measured by height and weight in units of $Kg/m^2$, Impedance BMI measured in increments of 0.1 in units of $Kg/m^2$, 4pc (4 principle components to account for ancestry).

## BMI

Body mass index (BMI) has been shown to increase SARS-CoV-2 risk across many populations [3]. Interestingly, we found that BMI was associated with an increased odds of SARS-CoV-2 positive testing (OR = 1.12, 95% CI = 1.06–1.18, p-value = $6.14 \times 10^{-5}$; Table 2) but when HDL was controlled for this effect was no longer significant (OR = 1.06, 95% CI = 0.995–1.13, p-value = 0.071; Table 2). These findings were consistent when BMI was measured by electrical impedance (OR = 1.12, 95% CI = 1.06–1.18, p-value = $8.54 \times 10^{-5}$; Table 2) and the significance was also lost when HDL-C was controlled for (OR = 1.06, 95% CI = 0.994–1.13, p-value = 0.077; Table 2).

## HDL-cholesterol/ApoA-I

After controlling for age, sex and 4pc, we found that plasma HDL-C levels were associated with a reduced odds of testing positive for SARS-CoV-2 (OR = 0.845, 95% CI = 0.788–0.907, p-value = $2.45 \times 10^{-6}$; Table 3), this effect was maintained even when controlling for BMI (OR = 0.863, 95% CI = 0.801–0.93, p-value = $1.17 \times 10^{-4}$; Table 3). Moreover, we found that plasma levels of Apolipoprotein A-I (ApoA-I), the major protein component of HDL-C particles in plasma, were also associated with a reduced odds of testing positive for SARS-CoV-2 (OR = 0.849, 95% CI = 0.793–0.910, p-value = $2.90 \times 10^{-6}$; Table 3). The effect of ApoA-I also remained significant when controlling for BMI (OR = 0.865, 95% CI = 0.806–0.929, p-value = $6.97 \times 10^{-5}$; Table 3). Consistent with high collinearity between HDL-C and ApoA-I, the effect of either is negligible when the opposite is controlled for suggesting that both describe the same effect (Table 3).

**Table 3. Effect of HDL and ApoA.**

| Trait | Covariates | N | Cases/Controls | Mean (S.D.) | Odds Ratio | 95% Confidence Interval | p-value |
|---|---|---|---|---|---|---|---|
| HDL-C | Age, Sex, 4pc | 7,821 | 1,291/6,530 | 1.40 (0.38) | 0.845 | 0.788–0.907 | **$2.45 \times 10^{-6}$** |
| | Age, Sex, 4pc, ApoA-I | 7,777 | 1,286/6,491 | 1.39 (0.37) | 0.944 | 0.804–1.11 | 0.480 |
| | Age, Sex, 4pc, BMI | 7,764 | 1,280/6,484 | 1.40 (0.38) | 0.863 | 0.801–0.93 | **$1.17 \times 10^{-4}$** |
| ApoA-I | Age, Sex, 4pc | 7,783 | 1,288/6,495 | 1.51 (0.27) | 0.849 | 0.793–0.910 | **$2.90 \times 10^{-6}$** |
| | Age, Sex, 4pc, HDL-C | 7,777 | 1,286/6,491 | 1.51 (0.27) | 0.894 | 0.762–1.048 | 0.167 |
| | Age, Sex, 4pc, BMI | 7,727 | 1,277/6,450 | 1.51 (0.27) | 0.865 | 0.806–0.929 | **$6.97 \times 10^{-5}$** |

*High density lipoprotein (HDL-C) measured in mmol/L, Apolipoprotein A-I (ApoA-I) measured in g/L.

**Table 4. Effect of hyperlipidemia, LDL, ApoB and triglycerides.**

| Trait | Covariates | N | Cases/Controls | Mean (S.D.) | Odds Ratio | 95% Confidence Interval | p-value |
|---|---|---|---|---|---|---|---|
| Hyperlipidemia | Age, Sex, 4pc | 7,515 | 1,277/6,238 | N/A | 1.351 | 1.010–1.807 | **0.043** |
| | Age, Sex, 4pc, HDL-C | 6,522 | 1,083/5,439 | N/A | 1.261 | 0.916–1.736 | 0.155 |
| | Age, Sex, 4pc, ApoA-I | 6,489 | 1,081/5,408 | N/A | 1.273 | 0.924–1.753 | 0.140 |
| | Age, Sex, 4pc, BMI | 5,957 | 1,267/6,195 | N/A | 1.259 | 0.937–1.69 | 0.126 |
| LDL-C | Age, Sex, 4pc | 8,532 | 1,413/7,119 | 3.44 (0.89) | 0.995 | 0.939–1.055 | 0.872 |
| ApoB | Age, Sex, 4pc | 8,498 | 1,404/7,094 | 1.01 (0.24) | 1.003 | 0.947–1.063 | 0.910 |
| Triglycerides | Age, Sex, 4pc | 8,534 | 1,411/7,123 | 1.81 (1.06) | 1.026 | 0.969–1.087 | 0.375 |

*Low density lipoprotein (LDL-C) measured in mmol/L, Apolipoprotein B (ApoB) measured in g/L, triglycerides measured in mmol/L.

## Hyperlipidemia/LDL-cholesterol/ApoB/triglycerides

The diagnosis of hyperlipidemia (ICD10 codes E78.4 and E78.5) was modestly associated with an elevated odds of testing positive for SARS-CoV-2 (OR = 1.362, 95% CI = 1.021–1.817, p-value 0.036; Table 4). However, when ApoA-I, HDL-C or BMI were controlled for, this effect was no longer significant (Table 4). In contrast to prior studies, we did not find any association between LDL-C levels and odds of testing positive for SARS-CoV-2 (OR = 0.995, 95% CI = 0.939–1.055, p-value = 0.872; Table 4) [9]. Consistent with this, apolipoprotein B, the primary lipoprotein associated with plasma LDL-C, was not associated with any significant effect on odds of testing positive for SARS-CoV-2 (OR = 1.003, 95% CI = 0.947–1.063, p-value = 0.910; Table 4). Additionally, no significant effect of triglyceride levels on odds of testing positive for SARS-CoV-2 were found (OR = 1.026, 95% CI = 0.969–1.087, p-value = 0.375; Table 4).

## Diabetes/HbA1c

The diagnosis of Type II diabetes (Phecode phe250.2) was associated with an elevated odds of testing positive for SARS-CoV-2 (OR = 1.213, 95% CI = 1.028–1.432, p-value = 0.0225; Table 5). Consistent with this, we found that HbA1c level was associated with a significant increase in odds of testing positive for SARS-CoV-2 (OR = 1.061, 95% CI = 1.005–1.121, p-value = 0.0332; Table 5). However, when ApoA-I, HDL-C or BMI were controlled for, the effects of both the type II diabetes diagnosis and HgA1c level were no longer significant

**Table 5. Effect of diabetes and HbA1c.**

| Trait | Covariates | N | Cases/Controls | Mean (S.D.) | Odds Ratio | 95% Confidence Interval | p-value |
|---|---|---|---|---|---|---|---|
| Type II Diabetes | Age, Sex, 4pc | 8,948 | 1,500/7,448 | N/A | 1.213 | 1.028–1.432 | **0.023** |
| | Age, Sex, 4pc, HDL-C | 7,770 | 1,283/6,487 | N/A | 1.131 | 0.943–1.356 | 0.184 |
| | Age, Sex, 4pc, BMI | 8,882 | 1,488/7,394 | N/A | 1.126 | 0.947–1.34 | 0.180 |
| HbA1c* | Age, Sex, 4pc | 8,508 | 1,421/7,087 | 37.32 (8.43) | 1.061 | 1.005–1.121 | **0.033** |
| | Age, Sex, 4pc, HDL-C | 7,398 | 1,217/6,181 | 37.39 (8.51) | 1.035 | 0.974–1.100 | 0.265 |
| | Age, Sex, 4pc, BMI | 8,444 | 1,409/7,035 | 37.30 (8.41) | 1.03 | 0.972–1.09 | 0.317 |
| Type I Diabetes | Age, Sex, 4pc | 8,006 | 1,311/6,695 | N/A | 0.817 | 0.529–1.261 | 0.360 |

*HbA1c measured in mmol/mol.

(Table 5). Curiously, Type I diabetes (Phecode phe250.1) was not associated with an elevated odds of testing positive for SARS-CoV-2 (Table 5).

## Discussion

Early studies of SARS-CoV-2 pandemic identified pre-existing cardiovascular disease and obesity as risk factors associated with acquiring SARS-CoV-2 [2–8]. In addition prior studies found that hypolipidemia and declining HDL-C in the setting of acute Covid-19 illness was associated with disease severity [9, 10]. Moreover, obesity was associated with higher prevalence of SARS-CoV-2 infection and Covid-19 disease severity [3, 7, 8]. Here, we sought to validate these findings by examining the potential effects of baseline BMI, lipoproteins, their respective apolipoproteins and diabetes on the odds of acquiring SARS-CoV-2 in a cohort from the UK Biobank dataset. We confirmed that baseline hyperlipidemia was associated with the odds of testing positive for SARS-CoV-2, but this association was driven primarily by association of baseline lower HDL-C and ApoA-I levels with SARS-CoV-2 positivity, suggesting specifically that baseline HDL-C level may be useful for stratifying SARS-CoV-2 infection risk.

Moreover, consistent with an earlier study, we confirmed the association of high BMI with the odds of testing positive for SARS-CoV-2 [7]. Importantly, this effect was no longer significant when baseline HDL-C and ApoA-I levels were controlled for. We also found association of Type II (but not Type I) Diabetes and HbA1c levels with SARS-CoV-2 diagnosis. Again, when baseline ApoA-I or HDL-C levels were controlled for, the effects of both the Type II diabetes diagnosis and HbA1c level were no longer significant. Given that Type II diabetes is associated with reduced HDL-C levels, we hypothesize that the elevated odds associated with Type II diabetes is mediated in part through its effect on HDL-C levels. Taken together, our results suggest that HDL-C may be mediating part of the well-known effect of BMI on SARS-CoV-2 /Covid-19 risk and plays a greater role in SARS-CoV-2 pathogenesis than previously appreciated.

Although the findings of this study persist when appropriate controls are applied, we acknowledge the inherent limitations of this association study, which is subject to sampling bias. Importantly, we do not know the context in which the SARS-CoV-2 testing was carried out, the HDL-C status at the time of testing, and the disease severity of each case. Participants tested in this study were primarily those who presented to a clinical care site with symptoms suggestive of SARS-CoV-2. Although this has the potential to marginally increase SARS-CoV-2 positive testing rate, it is unlikely to influence the association of HDL-C, BMI and alcohol consumption with SARS-CoV-2 positivity rate. Additionally, while majority of those tested were inpatient at the time of sampling, we acknowledge the potential confounding effects of subsequent expansion of testing into the community and to asymptomatic patients; however, we believe that such effects would tend to diminish any association with SARS-CoV-2 test positive rates.

Another limitation of an association study based on UK Biobank is that the baseline cardiometabolic data was collected several years prior to the SARS-CoV-2 pandemic. However, we do not believe that this did not have a substantial impact on our findings. While mean BMI has been reported to have increased around the world from 1976 to 2016, this increase has plateaued in recent years in high-income English-speaking countries, including the UK [17]. Additionally, mean triglyceride, HDL-C and LDL-C levels tend to change only modestly or remain relatively stable in the population over time [18, 19]. We believe that these modest changes over time would tend to diminish their associations with SARS-CoV-2 test positive rates, particularly if abnormal baseline levels were treated in the interim with medications. While the concurrent use of medications at the time of SARS-CoV-2 infection among the

UKB participants is unknown, we examined the use of common cardiometabolic medications at the time of enrollment and found no association with the SARS-CoV-2 infection, except for metformin, whose significance disappeared when adjusted for HDL-C levels or BMI (S1 Data). Too few subjects (<1% of UKB participants) were on either niacin or a cholesteryl ester transfer protein (CEPT) inhibitor, which are known to raise HDL-C levels, to determine association with SARS-CoV-2 infection.

Despite these limitations, we believe that our exploration of SARS-CoV-2 corroborates some of the earlier studies and provides valuable insight and guidance for future studies. One of the most compelling finding of this study is the association of lower baseline HDL-C levels with SARS-CoV-2 positivity, which corroborates an earlier study, which found association of declining HDL-C levels with Covid-19 disease severity [10]. While causal inferences are beyond the scope of this study, and the potential mechanism by which HDL-C may confer protection from SARS-CoV-2 is unknown, given HDL-C's pleiotropic characteristics including antioxidant, antithrombotic, microvascular-protective, anti-apoptotic, and anti- as well as pro-inflammatory properties, it is plausible that HDL-C may play a protective role in preventing the establishment of SARS-CoV-2 infection [20–22]. Indeed, based on the protective effects of HDL-C/ApoA-I replacement strategies and cholesteryl ester transfer protein (CEPT) inhibition in septic shock, there is increasing recognition of the role of HDL-C in infection control, including the direct antiviral effects of HDL-C [21–24]. Our analysis opens up several avenues for further study, for example, to determine whether baseline HDL-C levels can identify high risk patients, to explore whether administration of CEPT inhibitors such as ezetimibe to elevate HDL-C levels may confer protection against SARS-CoV-2 infection, or even to examine whether HDL-C particle confers direct protection against SARS-CoV-2 infection.

## Supporting information

**S1 Data.**
(XLSX)

## Acknowledgments

This research was conducted using data from UK Biobank, a major biomedical database. www.ukbiobank.ac.uk.

## Author Contributions

**Conceptualization:** Ryan J. Scalsky, Jeffery R. O'Connell, James A. Perry, Charles C. Hong.

**Data curation:** Ryan J. Scalsky, Yi-Ju Chen, Karan Desai, Jeffery R. O'Connell, James A. Perry, Charles C. Hong.

**Formal analysis:** Ryan J. Scalsky, Yi-Ju Chen, Karan Desai, Jeffery R. O'Connell, James A. Perry, Charles C. Hong.

**Funding acquisition:** Charles C. Hong.

**Investigation:** Ryan J. Scalsky, Yi-Ju Chen, Jeffery R. O'Connell, James A. Perry, Charles C. Hong.

**Methodology:** Ryan J. Scalsky, Yi-Ju Chen, Jeffery R. O'Connell, James A. Perry, Charles C. Hong.

**Project administration:** Charles C. Hong.

**Supervision:** James A. Perry, Charles C. Hong.

**Writing – original draft:** Ryan J. Scalsky, Karan Desai, Jeffery R. O'Connell, James A. Perry, Charles C. Hong.

**Writing – review & editing:** Ryan J. Scalsky, James A. Perry, Charles C. Hong.

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
