## [Decision Letter · Decision Letter 0]

23 Dec 2020

PONE-D-20-36368

Baseline Cardiometabolic Profiles and SARS-CoV-2 Infection in the UK Biobank

PLOS ONE

Dear Dr. Hong,

Thank you for submitting your manuscript to PLOS ONE. After careful consideration, we feel that it has merit but does not fully meet PLOS ONE’s publication criteria as it currently stands. Therefore, we invite you to submit a revised version of the manuscript that addresses the points raised during the review process.

Please answer to all the criticisms raised by the Reviwers.

We look forward to receiving your revised manuscript.

Kind regards,

Laura Calabresi

Academic Editor

PLOS ONE

Journal Requirements:

2. In your ethics statement in the Methods section and in the online submission form, please provide additional information about the data used in your retrospective study.

Specifically, please ensure that you have discussed whether all data were fully anonymized before you accessed them and/or whether the IRB or ethics committee waived the requirement for informed consent.

If patients provided informed written consent to have data from their medical records used in research, please include this information.

3. Please clarify if the biological samples used in your study were:

(a) specifically collected for this study or not

(b) whether the samples were completely de-identified before researchers accessed the samples.

'This research was conducted using the UK Biobank Resource under Application Number 49852. This work was supported by 5T32GM092237-10 to RS, NIGMS R01GM118557, NHLBI R01HL135129 to CCH, and NHLBI 1U01HL137181 to JP. The funders had no role in the design and conduct of the study; collection, management, analysis and interpretation of the data; preparation, review or approval of the manuscript; or decision to submit the manuscript for publication.'

'No. The funders had no role in study design, data collection and analysis, decision to publish, or preparation of the manuscript.'

Reviewers' comments:

Reviewer's Responses to Questions

**Comments to the Author**

1. Is the manuscript technically sound, and do the data support the conclusions?

Reviewer #1: Yes

Reviewer #2: No

2. Has the statistical analysis been performed appropriately and rigorously? 

Reviewer #1: Yes

Reviewer #2: I Don't Know

3. Have the authors made all data underlying the findings in their manuscript fully available?

Reviewer #1: Yes

Reviewer #2: Yes

4. Is the manuscript presented in an intelligible fashion and written in standard English?

Reviewer #1: Yes

Reviewer #2: No

5. Review Comments to the Author

Reviewer #1: The authors tried to describe the impact of BMI, HDL-C, LDL-C, apoA-I, apoB, triglycerides, hemoglobin A1c, diabetes, alcohol and red wine intake on the odds of testing positive for SARS-CoV-2, taking advantage of the UK Biobank database. HDL-C and apoA-I were associated with decreased SARS-CoV-2 odds, while higher BMI and diabetes were associated with increased odds. However, the latter lost significance when controlled for HDL-cholesterol. The study design was appropriate.

Some minor comments:

- Please refer to HDL and LDL as HDL-cholesterol and LDL-cholesterol, which are more appropriate since you are measuring cholesterol and not the particles themselves.

- Please refer to apoA as apoA-I, since different apoA exist, with different biological roles

- Check abbreviations. They should be defined the first time they appear

- It would be useful to include a more detailed characterization of the included patients. Since lipids and other biochemical parameter were collected at the time of enrolment in the database, lipid-lowering treatment and hypoglycemic medications should be included in the description of patients and in the analysis.

- If available, association of these parameters with disease severity would be of interest

Reviewer #2: Besides pharmacological approaches, the outbreak COVID-19 has led the scientific community to find biomarker that could have help to identify people who are at the risk the most. Although the present article is aimed at identifying which metabolic biomarker could be more suitable to identify these individuals, the manuscript in this form seems to fail in some statements, e.g., which is the portrait linking red wine and COVID-19? More specifically, which is the scientific path bonding drinking intake and all the other parameters? I would leave out this parameter. Instead, which is the role of insulin?

The aims should be described in a more detailed way.

Overall, as stated in the limitations, these parameters have been collected several years prior the outbreak and thus no recent records exist. Lipoproteins are closely related to infections and recent data on COVID-19 have correlated lipoproteins with outcomes (death) instead of odds of testing positive for SARS-CoV-2. Which is a plausible explanation?

The discussion should address mechanist pathways related to the parameters that have been analyzed.

6. PLOS authors have the option to publish the peer review history of their article (what does this mean?). If published, this will include your full peer review and any attached files.

Reviewer #1: No

Reviewer #2: No

---

## [Author Response · Author response to Decision Letter 0]

5 Jan 2021

Dear Dr. Calabresi

Enclosed is our revised manuscript “Baseline cardiometabolic profiles and SARS-CoV-2 infection in the UK Biobank,” which we would like to resubmit for publication in PLoS ONE .

We thank the Reviewers for their expert comments on our original submission, and have made the appropriate revisions. Our response to specific comments are italicized in red. 

Reviewer #1:

• Please refer to HDL and LDL as HDL-cholesterol and LDL-cholesterol, which are more appropriate since you are measuring cholesterol and not the particles themselves. We have made the corrections as requested.

• Please refer to apoA as apoA-I, since different apoA exist, with different biological roles. We have made the corrections as requested.

• Check abbreviations. They should be defined the first time they appear. We have made the corrections as requested.

• It would be useful to include a more detailed characterization of the included patients. Since lipids and other biochemical parameter were collected at the time of enrolment in the database, lipid-lowering treatment and hypoglycemic medications should be included in the description of patients and in the analysis. We have included the analysis of the impact of lipid-lowering treatment and diabetes medications at the time of enrollment as Supplementary Data and dissuss them in the discussion. Briefly, while the concurrent use of medications at the time of SARS-CoV-2 infection among the UKB participants is unknown, we examined the use of common cardiometabolic medications at the time of enrollment and found no association with the SARS-CoV-2 infection, except for metformin, whose significance disappeared when adjusted for HDL-C levels or BMI.

• If available, association of these parameters with disease severity would be of interest. Disease severity data is not available from the UK Biobank.

Reviewer #2

• Besides pharmacological approaches, the outbreak COVID-19 has led the scientific community to find biomarker that could have help to identify people who are at the risk the most. Although the present article is aimed at identifying which metabolic biomarker could be more suitable to identify these individuals, the manuscript in this form seems to fail in some statements, e.g., which is the portrait linking red wine and COVID-19? More specifically, which is the scientific path bonding drinking intake and all the other parameters? I would leave out this parameter. Instead, which is the role of insulin? We agree with the Reviewer 2; therefore, we have removed the section on red wine intake. We examine the impact of insulin on SARS-CoV-2 infection and found no association. This information is now included in the Data Supplement.

• The aims should be described in a more detailed way. We have modified the objective in the abstract to specifically state that “this study seeks to examine the impact of additional baseline cardiometabolic risk factors including high density lipoprotein-cholesterol (HDL-C), low density lipoprotein-cholesterol (LDL-C), Apolipoprotein A-I (ApoA-I), Apolipoprotein B (ApoB), triglycerides, hemoglobin A1c (HbA1c) and diabetes on the odds of testing positive for SARS-CoV-2 in UK Biobank (UKB) study participants.” 

• Overall, as stated in the limitations, these parameters have been collected several years prior the outbreak and thus no recent records exist. Lipoproteins are closely related to infections and recent data on COVID-19 have correlated lipoproteins with outcomes (death) instead of odds of testing positive for SARS-CoV-2. Which is a plausible explanation? The discussion should address mechanist pathways related to the parameters that have been analyzed. We agree with the reviewer #2 and have expanded discourse on these topics in the Discussions. To paraphrase, while the present study only presents associations between HDL-C and SARS-CoV-2 infectivity, our findings corroborate an earlier study, which found association of declining HDL-C levels with sepsis in general and Covid-19 disease severity in particular. Together with the emerging data on the protective effects of HDL-C/ApoA-I replacement strategies and cholesteryl ester transfer protein (CEPT) inhibition in septic shock, and increasing recognition of the role of HDL-C in infection control, including the direct antiviral effects of HDL-C, we believe that our study highlights mechanistic and therapeutic insights that deserves further attention. 

We trust that the reviewers and the editors will find the revise manuscript much improved and is worthy of publication in PLoS ONE.

Sincerely yours,

Charles C. Hong, MD, PhD

University of Maryland School of Medicine

Baltimore, MD

---

## [Decision Letter · Decision Letter 1]

2 Mar 2021

Baseline cardiometabolic profiles and SARS-CoV-2 infection in the UK Biobank

PONE-D-20-36368R1

Dear Dr. Hong,

We’re pleased to inform you that your manuscript has been judged scientifically suitable for publication and will be formally accepted for publication once it meets all outstanding technical requirements.

Kind regards,

Laura Calabresi

Academic Editor

PLOS ONE

Additional Editor Comments (optional):

Reviewers' comments:

Reviewer's Responses to Questions

**Comments to the Author**

1. If the authors have adequately addressed your comments raised in a previous round of review and you feel that this manuscript is now acceptable for publication, you may indicate that here to bypass the “Comments to the Author” section, enter your conflict of interest statement in the “Confidential to Editor” section, and submit your "Accept" recommendation.

Reviewer #1: All comments have been addressed

Reviewer #2: All comments have been addressed

2. Is the manuscript technically sound, and do the data support the conclusions?

Reviewer #1: Yes

Reviewer #2: Yes

3. Has the statistical analysis been performed appropriately and rigorously? 

Reviewer #1: Yes

Reviewer #2: Yes

4. Have the authors made all data underlying the findings in their manuscript fully available?

Reviewer #1: Yes

Reviewer #2: Yes

5. Is the manuscript presented in an intelligible fashion and written in standard English?

Reviewer #1: Yes

Reviewer #2: Yes

6. Review Comments to the Author

Reviewer #1: (No Response)

Reviewer #2: (No Response)

7. PLOS authors have the option to publish the peer review history of their article (what does this mean?). If published, this will include your full peer review and any attached files.

Reviewer #1: No

Reviewer #2: No

---

## [Editor Report · Acceptance letter]

23 Mar 2021

PONE-D-20-36368R1 

Baseline cardiometabolic profiles and SARS-CoV-2 infection in the UK Biobank 

Dear Dr. Hong:

I'm pleased to inform you that your manuscript has been deemed suitable for publication in PLOS ONE. Congratulations! Your manuscript is now with our production department. 

Kind regards, 

on behalf of

Prof. Laura Calabresi 

Academic Editor

PLOS ONE